# Mechanisms of Photoreceptor Death in Retinitis Pigmentosa

**DOI:** 10.3390/genes11101120

**Published:** 2020-09-24

**Authors:** Fay Newton, Roly Megaw

**Affiliations:** 1MRC Human Genetics Unit, University of Edinburgh, South Bridge, Edinburgh EH8 9YL, UK; roly.megaw@ed.ac.uk; 2Princess Alexandra Eye Pavilion, NHS Lothian, Edinburgh EH3 9HA, UK

**Keywords:** retinitis pigmentosa, photoreceptor, inherited retinal disease, apoptosis, regulated necrosis, autophagy, microglia, clinical trials

## Abstract

Retinitis pigmentosa (RP) is the most common cause of inherited blindness and is characterised by the progressive loss of retinal photoreceptors. However, RP is a highly heterogeneous disease and, while much progress has been made in developing gene replacement and gene editing treatments for RP, it is also necessary to develop treatments that are applicable to all causative mutations. Further understanding of the mechanisms leading to photoreceptor death is essential for the development of these treatments. Recent work has therefore focused on the role of apoptotic and non-apoptotic cell death pathways in RP and the various mechanisms that trigger these pathways in degenerating photoreceptors. In particular, several recent studies have begun to elucidate the role of microglia and innate immune response in the progression of RP. Here, we discuss some of the recent progress in understanding mechanisms of rod and cone photoreceptor death in RP and summarise recent clinical trials targeting these pathways.

## 1. Introduction

Retinitis pigmentosa (RP) is an inherited retinal dystrophy affecting 1 in 3000 people globally [1]. Mutations in one of over 150 causal genes identified to date leads to the classical clinical appearance of bone spicule formation in the peripheral retina, blood vessel attenuation and optic disc pallor. The disease is characterised by the progressive degeneration and death of rod photoreceptors, resulting in an initial loss of night vision (nyctalopia) and a progressive constriction of the visual field. Once rod cells are severely depleted, non-autonomous death of cones results, causing severe central visual loss [1,2]. 

RP can be inherited in an autosomal dominant, autosomal recessive, X-linked or mitochondrial manner and can be either syndromic or non-syndromic [2,3]. Genes have been identified that are involved in phototransduction, cell trafficking and rhodopsin recycling pathways [3,4,5]. However, despite knowledge of how these mutations disrupt photoreceptor function, the precise mechanisms that lead to photoreceptor death are still not well understood. Given the extremely heterogeneous nature of RP, identifying and understanding common cell death mechanisms is vital if we are to develop treatments applicable to all disease-causing alleles. This review brings together some of the recent work to identify cell death pathways involved in RP and the mechanisms that link causative mutations to initiation of these pathways. 

## 2. Cell Death Mechanisms

### 2.1. Apoptosis

Apoptosis is the most well studied form of programmed cell death and has long been considered to be the predominant mechanism in neurodegenerative diseases, including retinal degeneration [6,7,8]. Crucially, apoptosis is characterised by the sequestering of cellular components and removal of cell debris, thus minimising damage to the surrounding tissue. Initiation of apoptosis depends on the activation of caspase proteins. Intrinsic activation of apoptosis through detection of cell damage or loss of survival signals leads to increased expression of pro-apoptotic proteins such as Bax, resulting in the release of mitochondrial proteins, including cytochrome c [9]. Cytoplasmic cytochrome c forms a complex with Apoptotic Protease Activating Factor 1 (APAF1) and caspase-9, termed the apoptosome. Activated caspase-9 in this complex then cleaves and activates executioner caspases (caspase-3, 6 and 7) [9,10]. This results in a cascade of events ending in DNA and protein fragmentation, chromatin and cytoskeleton condensation and the formation of apoptotic bodies (see Figure 1B). Apoptosis can also be activated extrinsically following interaction between ligands produced by immune cells and death receptors (such as Tumour Necrosis Factor (TNF) or Fas) on the surface of damaged cells. Ligand-bound receptors recruit and activate caspase-8 and this ultimately leads to activation of executioner caspases and apoptotic body formation [10]. 

### 2.2. Regulated Necrosis

Although photoreceptor death by apoptosis has been reported in models of retinal degeneration [7,8,11,12], there is growing evidence that caspase-independent cell death mechanisms are also involved in many blinding diseases, including RP. Indeed, some recent studies have suggested that these alternative mechanisms may be the dominant modes of cell death in RP [13,14]. Several forms of regulated necrosis have recently been described; including necroptosis, ferroptosis, pyroptosis and parthanatos [15,16]. The common outcome of all these mechanisms is the breakdown of the cytoplasmic membrane, and the resulting tissue damage elicits a strong inflammatory response. However, different molecular pathways are involved in their initiation. 

Necroptosis is regulated by the interaction between two receptor interacting protein kinases (RIP1 and RIP3) [17,18]. This complex activates phosphorylation and oligomerisation of mixed lineage kinase domain-like protein (MLKL) and these oligomers form membrane pores, resulting in the loss of cell membrane integrity (Figure 1C) [19]. Initiation of this process can occur by activation of cell death receptors, toll-like receptors, interferons or DNA damage sensors. Necroptosis has been implicated in both rod and cone degeneration in animal models of RP in several recent studies [14,20,21,22], discussed further below. 

Ferroptosis is characterised by the accumulation of intracellular iron (Fe^2+^) and deregulation of glutamate metabolism, leading to increased lipid peroxidation and the build up of reactive oxygen species (ROS; Figure 1C) [23]. There is evidence that treatment of RP mouse models with iron chelators or over-expression of glutathione peroxidase 4 (GPX4; an inhibitor of lipid peroxidation) improves photoreceptor survival [24,25,26]. This suggests that ferroptosis may also contribute to cell death in some forms of RP. 

Pyroptosis may be considered an alternative form of apoptosis that, although dependent on caspase activation, results in cell membrane lysis and an inflammatory response. Pyroptosis occurs when gasdermin D or gasdermin E are cleaved by pro-inflammatory caspase-1 [27]. The N-terminal portions of these proteins insert into the cell membrane causing permeablisation and rupture. Although there has been no evidence presented so far showing pyroptosis of photoreceptors in RP, pyroptosis has been observed in damaged retinal pigment epithelium (RPE) cells following inhibition of the Thioredoxin (Trx/TrxR) redox system [28]. 

Another distinct cell death mechanism is parthanatos, a process that depends on poly-ADP-ribose polymerase (PARP) activity [29]. Although parthanatos does not require caspase activity, making it distinct from apoptosis, it does require nuclear translocation of the mitochondrial apoptosis-inducing factor (AIF). Nuclear translocation of AIF results in large-scale DNA fragmentation and chromatin condensation, leading to cell death (Figure 1C). As with other forms of regulated necrosis, parthanatos does not induce apoptotic bodies and instead results in the loss of cell membrane integrity. At least one recent study has observed increased PARP activation in mouse models of several known RP mutations, suggesting that parthanatos may contribute to photoreceptor death in many forms of RP [13]. 

### 2.3. Autophagy

Autophagy is a lysosome-mediated process that degenerates and recycles cell components. There are three main types of autophagy: macroautophagy, chaperone-mediated autophagy (CMA) and microautophagy. Macroautophagy involves cytoplasmic material being engulfed by autophagosomes, which then fuse with lysosomes, allowing this material to be degraded [30,31]. In CMA, proteins harbouring a specific amino acid sequence (consensus KFERQ) are bound by the Hsc70 chaperone and targeted to lysosomes via the lysosome-associated membrane glycoprotein 2A (LAMP-2A) receptor [32], while in microautophagy, material to be degraded enters lysosomes through invagination of the lysosomal or endosomal membrane (Figure 1D) [33]. Autophagy is important for mediating the degradation of damaged organelles and mis-folded or aggregated proteins. Cellular response to oxidative and metabolic stress can also activate autophagy to preserve cell and tissue homeostasis. However, when damage is severe, autophagy can lead to apoptotic cell death; indeed, many regulators of autophagy pathways also have central roles in regulating apoptosis [34]. Autophagy has been associated with photoreceptor death in RP models in many different studies [35,36,37,38], but the precise role of this pathway remains controversial. As discussed later in this review, increased autophagy flux (the dynamic process of autophagy) appears to be protective in some animal models of RP while other studies have suggested autophagy reduces photoreceptor survival [22,35,36]. 

## 3. Pathways Leading to the Death of Rod Photoreceptors in RP

Several mechanisms have been linked to triggering the death of rod photoreceptors in mouse models of RP via the pathways outlined above. These include the accumulation of mis-folded or mis-localised proteins, leading to endoplasmic reticulum (ER) stress, dysregulation of cGMP signalling, Ca^2+^ accumulation, oxidative stress and inflammatory responses. The causal mechanism of cell death depends on the genetic cause and often a combination of these processes leads to the activation of cell death pathways. 

### 3.1. Oxidative Stress

Photoreceptors have a high metabolism and high levels of mitochondrial function and oxygen consumption, making them vulnerable to oxidative stress (Figure 2A) [39]. Oxygen is the final acceptor of the electron transport chain in aerobic respiration; however, chance interactions between oxygen and upstream electron donors result in the generation of superoxide radicals, which can induce oxidative damage to lipids, protein and DNA within the cell. The formation of superoxides becomes more likely when tissue oxygen levels are high, as is the case in photoreceptors. The rich blood supply of the choroid maintains the retina in a highly oxygenated state [40] in order to facilitate the high metabolic demands of the visual cycle and allow for normal vision. As a result, the system risks tipping into hyperoxia, which has been shown to cause photoreceptor death in several studies [39,41]. Oxygen levels in the outer retina have been shown to increase in rat models of RP as degeneration progresses, due to decreased oxygen consumption following the death of a portion of rod cells [42,43]. The resulting exposure to high oxygen levels would be expected to result in increased oxidative stress in the remaining rods. However, activated microglia may also provide a source of superoxides in the outer retina. Up-regulation of NADPH-oxidase was observed in microglia that had migrated to the outer nuclear layer in the early stages of degeneration in *rd* mice [44]. This was accompanied by an increase in extracellular ROS in the outer nuclear layer. 

Treatment with antioxidants has been found to improve cell survival and preserve photoreceptor function in both N-methyl-N-nitrosurea (MNU) damaged and *rd10* (*Pde6b* mutant) mouse retinas [12,45]. Oxidative stress may induce apoptotic cell death, with the Trx redox system acting as a sensor in some systems [46]. In *rd10* mice, antioxidant treatment resulted in a reduction in inflammatory markers and p38 expression, suggesting that oxidative stress contributes to the induction of apoptosis via the NF-κB pathway [12] (Figure 2A). However, there is increasing evidence that oxidative stress can induce other forms of cell death, including necroptosis [47] and parthanatos [48]. It remains to be determined how these pathways might contribute to cell death following oxidative stress in RP. 

### 3.2. Metabolic Stress

The phosphoinositide 3 kinase/mammalian target of rapamycin (PI3K/mTOR) signalling pathway is involved in quiescence and cell survival and is linked to cancer, where its over-activation has an anti-apoptotic effect. Several studies have implicated the PI3K/mTOR pathway in photoreceptor death in RP, although its precise role is yet to be defined (Figure 2A). Up-regulation of the mTOR pathway by rod-specific ablation of the mTOR inhibitor Tsc1 was found to improve photoreceptor survival and visual function in *Pde6b* mutant mice [49]. Similarly, overexpression of S6 kinase B1 (S6K1), a downstream target of mTOR, in either rods or cones of *rd10* mice, improves cell survival and visual function [50]. In this study, *rd10* retinas were found to have considerably lower levels of activated of p-Akt and p-mTOR. Phosphatase and tensin homologue (PTEN), a negative regulator of the PI3K/Akt pathway, was up-regulated in *rd10* photoreceptors and the authors also showed that cell-specific deletion of PTEN reduced apoptosis of photoreceptors in *rd10* mice, leading to enhanced visual function. Further, inhibition of the PI3K/mTOR pathway has also been observed in MNU damaged photoreceptors in culture, leading to the activation of autophagy [38]. Prolonged autophagy activation resulted in the accumulation of autophagic vacuoles, indicated by enlarged lysosomes, and ultimately led to cell death by apoptosis. Down-regulation of mTOR signalling may therefore have a significant role in inducing photoreceptor death, at least for some genetic causes of RP. However, Li et al. also showed that inhibition of autophagy increased apoptosis in MNU damaged cells, accelerating the accumulation of ROS and activation of caspase-3. This implies that neither autophagy nor apoptosis is the sole mechanism of damage-induced cell death and that the balance between these pathways may be critical for photoreceptor survival.

### 3.3. ER Stress and Calcium Regulation

ER stress results from an imbalance between the demand for protein synthesis and the capacity of the ER for protein folding. This can be buffered by activation of the unfolded protein response (UPR), an adaptive process that reduces unfolded protein load in the ER and restores homeostasis [51]. The ER transmembrane protein kinase R-like endoplasmic reticulum kinase (PERK), inositol-requiring enzyme 1 (Ire1) and activating transcription factor 6 (ATF6) sense ER stress and initiate several signalling pathways that act in combination to inhibit protein synthesis and enhance degradation of mis-folded proteins (Figure 2B) [51]. However, if these responses fail to restore homeostasis, prolonged UPR can lead to cell death [52]. Excessive oligomerisation of Ire1α induces apoptosis by up-regulation of the c-Jun N-terminal kinase (JNK) pathway and pyroptosis via activation of the inflammasome [53,54]. Hyperactivation of PERK results in up-regulation of the ER stress marker C/EBP homologous protein (CHOP), leading to apoptosis by inhibition of anti-apoptotic B-cell lymphoma 2 (Bcl-2) and enhanced production of ROS [52,55]. 

ER stress caused by the accumulation of mutated proteins that cannot properly fold is thought to be one of the pathological mechanisms resulting from dominant, RP-causing rhodopsin mutations. However, the contribution of ER stress to cell death in RP is still not well understood. Increased *Chop* mRNA expression has been observed in *Rho^P23H/+^* transgenic rats and this correlated closely with the timing of photoreceptor degeneration [56], suggesting that ER stress triggers apoptosis (Figure 2B). More recently, inhibition of eukaryotic translation initiation factor 2α (eIF2α), a downstream effector of PERK, was found to reduce photoreceptor death in severe P23H transgenic (*P23H^Tg^*) and *Rho^P23H/−^* models but not in the milder *Rho^P23H/+^* model [57]. Similarly, this study also showed that activation of ATF4, (a pro-apoptotic transcription factor activated by ER stress signalling) increased cell death in *P23H^Tg^* and *Rho^P23H/−^* but not *Rho^P23H/+^*. ER stress may therefore have a role in eliciting photoreceptor death in circumstances where the level of mis-folded Rho exceeds a critical level, perhaps that of the wild type protein. However, the *Rho^P23H/+^* model may be considered a better representation of autosomal dominant RP.

Conversely, other studies have proposed that the ER stress response may have a protective function. Up-regulation of Ire1 signalling was observed in *Rho^P23H/+^* rats, leading to increased ER-associated protein degradation (ERAD), however pro-apoptotic signalling was not induced [58]. Ablation of *Chop* had no effect on retinal degeneration in these rats, suggesting that ER stress primarily results in the removal of mis-folded Rho via ERAD rather than inducing cell death. In support of this, another study found that inhibition of PERK in *Rho^P23H/+^* rats reduced photoreceptor function and survival and increased inclusion formation [59]. PERK signalling and the UPR may therefore act as an early defensive mechanism against the toxic effects of mis-folded Rho accumulation.

#### 3.3.1. cGMP Regulation of Calcium

ER stress responses can also arise from defects in the regulation of photoreceptor levels of cGMP [60], the second messenger whose signalling is critical for the phototransduction cascade. It has been hypothesised that elevated cGMP leads to increased Ca^2+^ influx via cyclic nucleotide-gated (CNG) channels [61,62], and that high intracellular Ca^2+^ can trigger ER stress responses [63]. Further, raised intracellular Ca^2+^ can activate a group of proteolytic enzymes known as Calpains, triggering apoptosis via the activation of AIF and caspase-7 [64] (Figure 1B,C). Whilst there is no direct evidence that raised intracellular calcium leads to ER stress-mediated photoreceptor death in RP, a body of work on phosphodiesterase subunit b (PDE6b)-mediated RP is suggestive of this.

Levels of cGMP are elevated in several animal models of RP with mutations in Pde6b [65]. More recently, increased plasma levels of cGMP were observed in a family with autosomal recessive RP carrying a homozygous mutation in *PDE6A* [66]. Additionally, recent transcriptome analysis of retinal organoids derived from patient-induced pluripotent stem cells with mutations in PDE6B has revealed changes in the expression of other genes related to the regulation of cGMP hydrolysis [67]. Loss of PDE6 function may therefore lead to photoreceptor degeneration due to inefficient hydrolysis of cGMP following phototransduction, with the resulting accumulation of Ca^2+^ causing ER stress-induced apoptosis. 

Knockdown of CNG channels, required for Ca^2+^ influx, slows photoreceptor degeneration in *Pde6g^−/−^* mice [68] despite elevated cGMP levels in *Cng1^−/−^Pde6g^−/−^* rods. This suggests that uncontrolled Ca^2+^ influx through open CNG channels may be responsible for initiating photoreceptor death in Pde6 mutants, rather than over-activation of other cGMP signalling pathways. However, this study did not measure calcium levels in *Pde6g^−/−^* or *Cng1^−/−^Pde6g^−/−^* rods. In contrast, induction of cGMP-dependent protein kinase (PKG) activity was found to be necessary and sufficient to trigger photoreceptor death in retinal explants in vitro, while inhibiting PKG activity in *rd1* (*Pde6b*) mutant mice improved photoreceptor survival [69]. Likewise, this study did not measure Ca^2+^ levels in either model. Conversely, Wang et al. found no effect of PKG knockdown on photoreceptor survival in *Pde6g* mutants. However, removal of PKG rescued retinal degeneration in mice lacking CNG channels (*Cngb1^−/−^Prkg1^−/−^*). This implies that excessive activation of PKG makes a greater contribution to the mechanism of cell death when CNG channels are absent, although again Ca^2+^ levels were not measured [68]. Elevated cGMP levels can therefore induce photoreceptor death by at least two different pathways, but CNG channels may be the primary target of elevated cGMP in *Pde6g^−/−^* photoreceptors. Neither of these studies analysed ER stress markers and it is therefore not clear whether ER stress is a factor in photoreceptor death downstream of cGMP accumulation in these models.

Conversely, reduced cGMP levels may activate cell death mechanisms through the induction of ER stress independently of Ca^2+^ accumulation. Mutations in the trafficking protein receptor expression enhancing protein 6 (REEP6) have been identified in several families with autosomal recessive RP [70]. *Reep6* mutant mice have severely reduced expression of guanylate cyclases GC1 and GC2 in rod cells, which convert guanosine triphosphate to cGMP [71]. Although cGMP levels were not measured in this study, the authors suggest that Reep6 may be required to maintain cGMP through stabilising or facilitating the trafficking of GCs. The loss of GC1 and GC2 from outer segments was accompanied by an increase in the expression of CHOP (an ER stress marker), the activation of caspase-12 (a marker of ER stress-induced apoptosis) [56,72], and changes in ER structure. However, since photoreceptor Ca^2+^ levels were not measured in this study, it is not clear whether intracellular Ca^2+^ is also depleted in *Reep6* mutants or if this is mitigated through enhanced Ca^2+^ release from the ER, as has been demonstrated in CNG channel-deficient cones [73].

#### 3.3.2. Calcium Regulates Apoptosis Independent of ER Stress

Several other studies have suggested that Ca^2+^ accumulation promotes photoreceptor death by caspase-dependent apoptosis. Enhanced intracellular Ca^2+^ has been observed in *rd1* photoreceptors in vitro, accompanied by increased calpain activation and nuclear translocation of AIF and caspase-12 (Figure 1B) [74]. This study also found that calcium channel blockers diminished calpain activation and apoptosis both in vitro and in *rd1* mice, suggesting that elevated Ca^2+^ is required for calpain activation and calpain-induced cell death. Subsequently, the pro-apoptotic factor Bax was found to be activated in three different models of RP (*rho^−/−^*, *rd1*, *Rho^P23H/+^*) [8]. Although this study did not measure calcium levels directly, they showed that the activation of Bax was regulated by calpain-1 and cathespin-D in *rd1* and *rho^−/−^* mice. However, the activation and mitochondrial localisation of Bax was only partially rescued by inhibiting these mechanisms in *Rho^P23H/+^* mutants. To investigate further, these researchers subsequently evaluated the different contributions of ER stress and calcium imbalance in *Rho^P23H/+^* mouse retinas [75]. The inhibition of calpains resulted in a strong reduction in caspase-7 activation and improved cell survival. However, reducing the ER stress response by inhibition of PERK did not affect caspase-7 activation and in fact accelerated retinal degeneration, supporting an earlier study that suggested that the activation of PERK has a protective effect [59]. They concluded that photoreceptor death by caspase 7-induced apoptosis is predominantly mediated by calpain activation rather than ER stress. 

#### 3.3.3. Calcium and Non-Apoptotic Cell Death

Interestingly, although the loss of caspase-7 was found to delay rod degeneration and apoptosis in mouse models of another dominant mutation (*Rho^T17M^*) [7,11], caspase-7 knockout had minimal effect in *rd1* or *rho^−/−^* mice [76]. This might suggest that calpain activation in *rd1* and *rho^−/−^* mice leads to a caspase-independent mechanism of cell death. Similarly, Ca^2+^ accumulation in rod cells accompanied by increased calpain-2 activation was observed in the early stages of photoreceptor degeneration in *rd10* mice [35]. The inhibition of calpain or cathepsin in these mice reduced cell death, however there was no increase in caspase activation in *rd10* retinas, suggesting that cell death does not occur by apoptosis.

An alternative model has been proposed, whereby increased cGMP leads to excessive protein phosphorylation by PKG in addition to high intracellular Ca^2+^, resulting in the activation of PARP as well as calpains (Figure 2B) [13]. This study observed increased activation of PARP in multiple mouse models of RP. However, they found no increase in Bax or caspase activation in any of the models tested, with the exception of the S334ter rhodopsin mutant, suggesting that apoptosis was not the primary cell death mechanism. They suggest that a parthanatos-like mechanism may be involved in photoreceptor death in at least some forms of RP. More recently, another group has observed up-regulation of RIP3 in rod cells in *Rho^P23H/+^* rats (Figure 1B), suggesting death by necroptosis [14]. Interestingly, this group did observe high expression of caspase-3 in S334ter rats, indicating that apoptotic pathways may also be important in some cases. 

### 3.4. Autophagy and Proteasome Regulation

The ubiquitin-proteasome system (UPS) is a complex of proteases responsible for the degradation of short lived proteins and soluble mis-folded proteins by proteolysis [77]. Autophagy is a lysosome-mediated process that can degrade or recycle long-lived proteins, insoluble protein aggregates and even whole organelles (Figure 1D). Autophagy is considered a dynamic process, with the term “autophagy flux” relating to the continued formation of first autophagosomes, followed by their fusion to lysosomes and finally their degradation [78]. Further, these two major intracellular protein degradation pathways (the UPS and autophagy) are interconnected, with a compensatory balance thought to exist between the two to ensure adequate clearance of cell debris [77]. Their dynamic relationship appears important for photoreceptor homeostasis, with an imbalance between the two being observed in several animal models (summarised in Figure 2B). 

A UPS–autophagy imbalance is apparent in autosomal dominant mutations, such as P23H, that lead to the mis-accumulation of mis-folded rhodopsin. *Rho^P23H/+^* mice have reduced proteasome function and increased autophagy activity, as well as increased expression of the UPR pathway regulator X-box binding protein 1 (XBP1) and the autophagy activator beclin1 [36]. The authors hypothesise that impaired proteasome function results in increased ER retention of accumulated mutant rhodopsin, leading to increased ER stress (evidenced by the up-regulation of XBP1) and resulting in autophagic cell death. They also observed increased proteasome subunits in autophagosomes from *Rho^P23H/+^* mice, suggesting that increased autophagy leads to further degradation of the proteasome. Indeed, drug-induced autophagy accelerates retinal degeneration in these mice. 

In contrast, reducing autophagy by rod-specific deletion of autophagy-related 5 (Atg5) (see Figure 1D) maintains photoreceptor structure and improves function in *Rho^P23H/+^* mutants. There is also an increase in proteasome activity, as evidenced by increased chymotrypsin-like activity in retinal lysates from *Rho^P23H/+^*Atg5^Δrod^ mice, and higher levels of proteasome 20S core and 19S regulatory subunits. Yao et al. therefore suggest that decreasing autophagy shifts degradation of mis-folded P23H rhodopsin towards the proteasome and this may reduce the ER retention of mutant protein, thus improving cell survival. In a subsequent study, this group showed that reducing protein mis-folding by treatment with the chaperone 4-PBA reduces ER stress and decreases autophagy activation, resulting in reduced degradation of proteasome subunits and hence restoring proteasome activity [37]. This was accompanied by improved visual function and reduced photoreceptor death. Similarly, increasing proteasome function by drug treatment reduces the shuttling of proteins to the autophagy pathway and improves photoreceptor survival [37]. This work argues that the strict control of autophagy flux, reducing degradation by this pathway, has a protective effect on stressed or damaged photoreceptors. 

Conversely, autophagy may have a protective function in the early stages of photoreceptor degeneration when other genes are involved, with other cell death mechanisms becoming up-regulated in later stages of RP as damage becomes more severe. For example, whilst autophagy is up-regulated in MNU-damaged photoreceptors in vitro, inhibition of autophagosome formation in these cells with 3-methyladenine accelerates cell damage and apoptosis [38]. Conversely, these authors found that prolonged activation of autophagy in MNU-damaged cells resulted in the accumulation of autophagic vacuoles, eventually leading to cell death by apoptosis. A similar biphasic role for autophagy has been observed in vivo. In *rd10* mice, the loss of the sigma-1 receptor (*rd10/SR1^−/−^*), a molecular chaperone that plays a protective role in neurodegenerative disease models, leads to ER stress [22,79]. Yang et al. showed that this is accompanied by increased levels of CHOP and the autophagy marker microtubule-associated protein light chain 3-II (LC3-II) in rods of three-week-old *rd10/SR1^−/−^* mice (see Figure 1D and Figure 2B), and these mice showed reduced photoreceptor degeneration compared to *rd10/SR1^+/+^* at this stage. However, the authors observed the opposite effect in older mice: levels of CHOP and LC3-II are reduced, accompanied by reduced photoreceptor function and increased activation of necroptosis [22]. Increasing autophagy in response to ER stress, therefore, could help to preserve photoreceptors initially, but this effect may be insufficient at later stages as other cell death mechanisms take over. 

Supporting this, an earlier study also found that LC3-II expression is reduced and autophagy flux is blocked prior to the onset of photoreceptor death in *rd10* mice [35]. This occurred in parallel with Ca^2+^ accumulation and calpain activation. The activation of proteolytic calpain enzymes can induce lysosomal membrane permeablisation (LMP) [80], leading to lysosomal damage, as well as cleavage of autophagy proteins and therefore a reduction in autophagy [81,82] (see Figure 2B). Rodríguez-Muela and colleagues showed that the inhibition of calpains reduces photoreceptor death and proposed that calpain-induced LMP was the mechanism responsible for inducing photoreceptor death in *rd10* mice. Interestingly however, they also observed that in the absence of calpain inhibition, down-regulation of autophagy pathways rescued photoreceptor death, while stimulating autophagy with rapamycin exacerbated photoreceptor loss, suggesting that autophagy may be damaging to cells with defective lysosomal function. 

The conflicting effects of autophagy in different mouse models of RP and at different stages of degeneration suggest that a balance between the UPS, autophagy, lysosomal function and other cell death mechanisms is needed to promote photoreceptor survival. Further work is required to fully define the roles of autophagy and the UPS in photoreceptor health and disease. 

## 4. Recruitment and Activation of Microglia and Monocytes

The inflammatory response and activation of innate immune cells also have a pivotal role in retinal degenerative diseases including RP [83]. Infiltration of degenerating regions of the retina by microglia and blood-derived macrophages has been observed in *rd1* and *rd10* mice and in post-mortem samples from RP patients [84,85,86]. Macrophages are mononuclear phagocytic cells that are capable of engulfing and destroying pathogens and damaged cells [87]. Blood-derived macrophages differentiate from circulating monocytes that infiltrate damaged or degenerating tissue in response to chemokine signals. In contrast, microglia are resident macrophages of the brain and retina and are required for local immune surveillance [87]. Microglia are derived from myeloid precursors during development, however their maintenance and local expansion within the retina depends entirely on the self-renewal of tissue-resident cells [88]. For clarity, monocyte-derived cells will be referred to as macrophages in this review while resident immune cells are referred to as microglia. In healthy retinas, resting microglia are confined to the ganglion cell layer, and the inner and outer plexiform layers, and have a ramified morphology (small cell body with long processes). However, during retinal degeneration, activation by cytokines results in the migration of microglia to the outer retina and morphological change to an amoeboid shape [89] (Figure 3). Another retinal glial cell type, Müller glia, may be important for activation and guidance of microglia and macrophages during retinal degeneration [90,91]. Müller glia are the most abundant macroglial cell type in the retina, providing homeostatic and metabolic support for photoreceptors and regulating synaptic activity in the inner retina [92]. The morphology of Müller cells, spanning the outer and inner retinal layers, facilitates transmission and response to signals from other retinal cells [92,93]. Müller glia may therefore coordinate the innate immune response to photoreceptor damage.

Innate immune response can be induced by the release of endogenous molecules from dying or degenerating photoreceptors; termed damage-associated molecular patterns (DAMPs). These include adenosine triphosphate (ATP), heat shock proteins (HSPs), DNA and RNA and are recognised by Toll-like receptors (TLRs), purigenic receptors, C-type leptin receptors and retinoic acid inducible gene 1 (RIG-I)-like receptors, resulting in the activation of an inflammatory response [94] (see Figure 3). The release of DAMPs triggers inflammasome activation in Müller glia and retinal microglia, leading to the activation and migration of microglia and blood-borne macrophages to the outer retinal layers and the release of chemokines, cytokines and other inflammatory factors from Müller glia and RPE cells [95,96]. Up-regulation of cytokines and infiltration of microglia/macrophages is associated with both apoptosis and regulated necrosis of photoreceptors in RP [20,97]. Activated microglia, macrophages and Müller glia also release inflammatory factors such as TNFα and can initiate apoptosis or necroptosis in other photoreceptors in the vicinity (Figure 3) [21,98]. However, the precise contributions of different innate immune cells and Müller glia to photoreceptor death in RP are still not well understood.

In many animal models of RP, increased infiltration of activated microglia, Müller glia and macrophages occurs before or during the early stages of rod degeneration [99,100,101,102]. Circulating monocyte-derived macrophages have been suggested to promote rod cell death in RP since ablation of Ccr2, the receptor for the monocyte chemoattractant MCP-1, reduced retinal degeneration in *rd10* mice [103]. However, a subsequent study suggested that monocyte-derived macrophages did not show significant infiltration of the outer nuclear layer and that resident (Ccr2-negative; [104]) microglia were responsible for most of the phagocytosis of photoreceptors in *rd10* mice [86]. Since many surface markers are shared between macrophages and retinal microglia, however, it is often difficult to distinguish between these cell types. Recent single cell RNA-seq studies of retinal degeneration models have attempted to address this [105,106]. Ronning et al. showed that monocytes, macrophages and microglia are present in degenerating retinas and identified distinct subsets of each of cell type [106]. Additionally, different subsets of retinal microglia occupy separate niches within the inner retina and may perform different functions following activation in disease states [105]. Functional heterogeneity of microglia and macrophages could therefore explain some of the conflicting findings regarding their role in photoreceptor death.

Pharmacological inhibition of microglial activation and cytokine release has been shown to reduce photoreceptor cell death [107,108]. Down-regulation of pathways involved in innate immunity and inflammation were also found to reduce retinal degeneration and preserve photoreceptor function in *Rho^P347S^* mice [109]. Interestingly, microglial activation has been shown to occur before the peak of rod photoreceptor apoptosis in *rd1* mice [100], and levels of inflammatory cytokines including IL-1β and TNFα increase before the onset of rod death [110]. Evidence has also been presented to suggest that phagocytosis can occur separately from apoptosis of photoreceptors and neurons [86,111]. Stressed but viable rods expose phosphatidylserine (PS) on the outer surface of the cytoplasmic membrane, marking them as targets for phagocytosis [86] (Figure 3). This study demonstrated that PS exposure occurred primarily on rods rather than cones in *rd10* retinas and on cells that were negative for caspase-3 and cleaved PARP. Microglia predominantly phagocytosed these living, non-apoptotic rods presenting surface PS, rather than dead or dying photoreceptors [86]. This suggests that microglia have an active role in retinal degeneration rather than simply a responsive role in the clearance of dying cells.

Müller glia may have an important role in the guidance of microglial and macrophage migration by secretion of chemotactic cues [91,112]. Müller glia have been shown to express most cytokines and inflammatory factors in vitro [113], and may therefore have a role in both the early response to photoreceptor degeneration and the activation of retinal microglia via the release of cytokines (Figure 3). Müller cells were found to increase secretion of the cytokine Cx3cl1 in an MNU-induced model of retinal degeneration, resulting in increased microglial activation, expression of Cx3cr1 receptor and migration to the outer retina [90]. This was accompanied by changes in microglial morphology, facilitating closer physical interaction with Müller cells. Depletion of Cx3cr1-positive cells was shown to reduce rod cell loss in *rd10* mice [86], suggesting that Cx3cr1-positive microglia are directly involved in photoreceptor death. Conversely, another study found that Cx3cr1-deficient *rd10* mice had higher microglial activation and infiltration [114]. This was accompanied by increased photoreceptor apoptosis and increased phagocytosis of photoreceptors by microglia. Cx3cl1–Cx3cr1 signalling may therefore have a regulatory role, ensuring efficient clearance of stressed photoreceptors. Microglia have also been shown to have a protective function in the retina in other studies [115,116,117]. Müller glia have also been shown to phagocytose dead and dying rods in *Rho^P23H/P23H^* mice, so they may also have a protective role [118]. The removal of stressed photoreceptors and debris from dying cells may reduce local inflammation and prevent further damage to neighbouring photoreceptors.

Activation of the complement pathway is also a feature of retinal degenerative diseases [119]. There are three separate complement-activating pathways (classical, lectin and alternative), however, all three result in a proteolytic cascade that leads to cleavage of component C3 on activating surfaces (e.g., pathogens or apoptotic cells). Cleavage of C3 produces the opsonin C3b, which facilitates phagocytosis, and an anaphylotoxin C3a [120,121] (Figure 3). Microglia express a range of complement genes and may also drive complement activation through the secretion of complement components [122,123]. The introduction of cytokines IL-6, IFNγ or TNFα to microglial cultures promotes the synthesis of complement components [122]. Increased C3 expression was observed in infiltrating microglia in the outer nuclear layer of *rd10* mice along with opsonisation of degenerating photoreceptors by C3b [124]. Interestingly, this study also showed that genetic ablation of C3 or the C3 receptor (CR3) accelerated photoreceptor degeneration. C3 or CR3-deficient microglia also showed decreased ability to phagocytose both apoptotic and living photoreceptors, but increased expression of inflammatory cytokines. This therefore has the dual effect of allowing the accumulation of apoptotic cells and increasing the neurotoxicity of microglia. The accumulation of dying photoreceptors may also directly impair the survival of nearby photoreceptors [125]. Silverman et al. therefore suggest that complement activation is an adaptive response that ensures the efficient removal of stressed or dying photoreceptors while limiting overall photoreceptor loss. 

## 5. Mechanisms of Secondary Cone Death in RP

Several theories have been proposed as to how the degeneration of rods eventually leads to cone cell death in later stages of RP. It has been suggested that dying rod cells might secrete a toxin that triggers cone death [126], however this does not explain the delay in cone loss. Cones are preserved until very late in the progression of RP when most rod cells have already been lost, whereas the peak of toxin secretion would be expected to occur much earlier, during the phase when most rods are dying. Alternatively, it has been proposed that cone death could be induced by activated microglia [84]. Primary death of rod photoreceptors leads to the activation of resident microglia [90,100], which then phagocytose degenerating rod cells [86]. It has been suggested that cytotoxic factors secreted by activated microglia kill adjacent photoreceptors, including cones, similar to the effects observed in CNS neurodegeneration in other diseases [84,127]. However, most microglial activation would occur during the period when rods are dying, so again this does not explain the delay in the onset of cone loss. 

It is also possible that the loss of rod-derived survival factors may be partially responsible for cone death and at least one such factor has been identified [128]. Nucleoredoxin-like 1 (Nxnl1) is alternatively spliced to produce two isoforms: truncated rod-derived cone viability factor (RdCVF) and full length RdCVF-Like (RdCVFL). RdCVF is secreted by rods and promotes cone survival. Injection of AAV–RdCVF in *rd10* and *Rho^P23H/+^* retinas resulted in improved cone function and delayed cone loss [129]. However, *Nxnl1^−/−^* mice do not exhibit a dramatic loss in cone density [130], suggesting that RdCVF is not essential for cone survival. In contrast, rod cells showed increased oxidative stress in *Nxnl1^−/−^* retinas, accompanied by the thinning of the outer nuclear layer. In agreement with this, Byrne et al. found that RdCVFL had little direct effect on cones, but reduced oxidative stress by-products in rods. It is possible that RdCVFL has a similar role to the anti-oxidant TRX1 in regulating redox homeostasis in rods, therefore protecting rod function and inhibiting apoptosis. It has also been suggested that the absence of RdCVF may make cones more susceptible to oxidative and metabolic stress [131,132]. RdCVF may therefore protect against cone cell death in RP, but it is unlikely to be the only mechanism.

Oxidative or metabolic stress remain the most likely mechanisms to explain the slow and irregular loss of cones observed in human RP patients. These pathways could both provide therapeutic targets to improve cone survival regardless of the underlying genetic cause. Rods cells have high oxygen consumption due to their high metabolic activity. As large numbers of rods die, oxygen consumption in the retina decreases. However, with an absence of autoregulation of blood vessels in the choroid, oxygen supply to the outer retina remains the same. This leads to excessive oxygen levels in degenerating retinas and, consequently, increased oxidative damage to cones. Antioxidant treatment has been shown to preserve cone function and slow cone death in both *rd10* and *Rho^Q334ter^* mouse models [133] and *Rho^P23H^* transgenic rats [134]. Cone degeneration is also more rapid in mice lacking SOD1, a component of the antioxidant defence system [26]. Treatment of *rd10* mice with a synthetic ligand of the sigma-1 receptor (SR1) was found to reduce oxidative stress and promote cone function and survival [135]. This effect was not seen in *rd10/SR1^−/−^* mice, suggesting that signalling through SR1 has a protective function. It is not clear, however, whether this is a direct effect of cones or a consequence of reducing oxidative stress in other cell types: SR1 is expressed on Müller glia and RPE cells as well as photoreceptors and, due to systemic administration of the SR1 ligand, all cells will have been targeted in this study. 

The PI3K/mTOR pathway may also function to protect cones following significant rod cell loss. Up-regulation of the mTOR pathway by conditional knockout of an mTOR inhibitor in *Pde6b* mutant rods was found to improve the survival of both rods and cones [49]. Constitutive activation of the mTOR complex has also been shown to slow the progression of cone death and preserve cone function in *rd1* and *Rho^−/−^* mice [136]. Activation of mTOR increased NADPH levels in cones and this study also found that cone loss was delayed in the absence of the NADPH-sensitive caspase-2, suggesting the mTOR pathway functions to reduce apoptosis. In an earlier study, Punzo et al. performed microarray analysis of RP mouse models at different time points during the progression of retinal degeneration. They found that genes associated with the insulin/mTOR pathway were significantly up-regulated during cone cell death [137]. These include some negative regulators of mTOR (*Grb10*, *Cab39*) but also some positive regulators (*Akt1*, *Hsp90*, *Rictor*), suggesting that mTOR may be up-regulated to protect dying cones. In support of this, they found that insulin treatment prolonged cone survival, while insulin depletion exacerbated cone cell death. Punzo et al. also observed increased autophagy in cones following significant loss of rod cells and observed high levels of LAMP-2 at the lysosomal membrane, suggesting increased CMA in dying cones (see Figure 1D). Since CMA can be induced by starvation, they suggested that gradual long-term starvation due to loss of rods (and hence loss of RPE-outer segment interactions required for shuttling nutrients to photoreceptors) may induce CMA and ultimately lead to cell death. The reduction in PI3K/mTOR activity by the overexpression of PTEN or siRNA knockdown of S6K1 has also been shown to induce apoptosis in a cone cell line in vitro [50]. This study also found that cone specific deletion of PTEN or overexpression of S6K1 in *rd10* mice reduced photoreceptor apoptosis. 

The mechanisms involved in the ultimate demise of cone photoreceptors are still not fully understood. While many of the studies detailed above suggest that cone death occurs by apoptosis, there is growing evidence for the involvement of other caspase-independent pathways. For example, it was shown that loss of caspase-7 does not affect cone survival in *rd1*, *Rho^−/−^* or *Rho^T17M^* models of RP [76]. Viringipurampeer and colleagues found that RIP3 dependent necroptosis appeared to be the primary mechanism of cone cell death in *Rho^S334ter^* rats [14]. In contrast, cone loss in *Rho^P23H^* retinas occurred through inflammasome activation. *Rho^P23H^* retinas exhibited higher levels of IL-1α and IL-1β than *Rho^S334ter^* retinas and the majority of microglia co-localised with cone arrestin. It is therefore possible that the final mechanism of cone cell death may not be entirely independent of the particular rod-derived mutation that drives RP.

## 6. Clinical Trials

Considerable progress has been made towards developing potential gene replacement, genome editing and stem cell treatments for RP and other retinal degenerative diseases [138,139,140,141]. However, such approaches are limited by the need to determine the specific underlying mutation (in the case of gene therapy) and the challenge of developing safe, reliable methods of administration with long-term efficacy and minimal adverse effects. These treatments will therefore take some time to reach clinic as they are approached allele by allele. Many recent studies have therefore focused on developing pharmacological approaches targeting cell death mechanisms as a means of delaying disease progression. Such treatments could also be used to supplement future gene or cell replacement therapies in order to improve treatment outcomes. Several compounds have been identified as having anti-apoptotic, anti-oxidant and anti-inflammatory effects in animal models of RP and some have shown promising preliminary results in clinical trials. Drugs targeting ER stress pathways and calcium channel inhibitors have also been explored. This section summarises some of the recent advances in developing these treatments. 

### 6.1. Neuroprotective Agents

Peptide growth factors and neurotrophic factors have been shown to inhibit apoptosis [142,143,144,145] and to protect against photoreceptor degeneration in animal models of RP [146,147,148]. There is some evidence for a beneficial effect of growth factors on preservation of photoreceptors in RP patients, at least in the short-term. In an attempt to deliver such factors to RP patients, Arslan et al. used subtenon injections of autologous platelet rich plasma (aPRP). PRP contains a high concentration of multiple growth factors including basic fibroblast growth factor (bFGF), vascular endothelial growth factor (VEGF), transforming growth factor (TGF-B1) and brain-derived neurotrophic factor (BDNF) along with pro- and anti-inflammatory cytokines [149,150]. However, as the PRP was not purified, the exact concentration of each component received by the patients is not certain. They found that aPRP improved visual field, microperimetry and multifocal electroretinography (ERG) values in patients nine weeks following treatment, although there was no significant effect on visual acuity [151]. Significant improvement in such a short time suggests that the treatment improved photoreceptor function rather than inhibited cell death, and the longer-term benefits and adverse effects remain unknown. In another study, short-term topical application of eye-drops containing nerve growth factor (NGF) purified from mouse submandibular glands was well tolerated in RP patients, without adverse effects. However, only a minority of the advanced-stage patients tested showed significant improvement in visual field after 30 days [152]. 

Ciliary neurotrophic factor (CNTF) is thought to protect against photoreceptor degeneration by up-regulating proteolysis inhibitors, reducing extracellular matrix degradation and complement activation [153]. However, the only long-term clinical trial involving early and late stages of RP reported that continuous release of CNTF from intravitreally injected implants led to a greater loss of visual field sensitivity than natural progression in sham-treated eyes [154]. Although this short-term visual loss was reversible, the study also found no evidence of long-term visual improvement after 60–96 months. 

Dietary antioxidant supplements such as vitamin A, Docosahexaenoic acid (DHA) and lutein are currently the only prescribed treatments for RP. However, there is still some debate as to their efficacy in slowing photoreceptor degeneration. Early randomised trials reported a protective effect of high doses of vitamin A [155]. However, these studies were not conducted in genotyped patients and there is some evidence that vitamin A supplement may not be suitable for all forms of RP. For example, vitamin A decreased the rate of retinal degeneration in the *Rho^T17M^* mouse model, but had no effect in *Rho^P347S^* mice [156]. It has also been suggested that a high intake of vitamin A could result in increased photoreceptor toxicity due to the production of high levels of all-trans retinal [157]. DHA is an omega-3 fatty acid and has been shown to have antioxidant and anti-inflammatory effects [158]. Combining DHA with vitamin A has been found to improve photoreceptor survival in RP in at least one clinical study [159]. A more recent study also reported that daily oral supplements of DHA slowed the progression of visual field loss in patients with X-linked RP over the course of a four-year study [160]. 

There has therefore been recent interest in other antioxidants as possible treatments for RP. Lycium barbaram polysaccharide (LBP), the main anti-oxidant component of Lycium barbaram, has been shown to be neuroprotective in various conditions [161] and may reduce cell death by increasing antioxidant defence and attenuating ROS production [162]. LBP was also recently reported to protect against photoreceptor degeneration via anti-oxidative, anti-inflammatory and anti-apoptotic mechanisms in *rd10* mice and MNU-induced rat models of retinal degeneration [163,164]. In a recent placebo-controlled clinical study, LBP was found to preserve macular structure and visual acuity in RP patients [165]. However, this study used a relatively small sample size and the genotypes of the RP subjects were not known, so it is not clear whether the beneficial effects would apply to all forms of RP. Subjects were only treated for 12 months, so the longer-term benefit remains to be determined. Nevertheless, while LBP may be a useful supplement for preserving visual function in RP patients, a larger scale longitudinal study using a defined dose of purified LBP would be needed to determine whether LBP has any role as a therapy in RP. Other experiments indicate that saffron may protect photoreceptors from oxidative damage in a rat model of light-induced photoreceptor degeneration [166] and attenuated rod and cone degeneration in *Rho^P23H^* rats [134]. The saffron-derived compounds safranal, crocetin and dimethylcrocetin have been shown to neutralise free radicals and protect DNA from oxidative damage in vitro and inhibit Bax up-regulation in rat neural (PC12) cells following acrylamide damage [167,168,169]. However, the precise mechanism of action in rat photoreceptors is not known. Although there have not yet been any clinical trials using saffron components to treat RP, recent trials have shown some promising early results for treatment of diabetic maculopathy and Stargardt Macular Dystrophy [170,171].

Inhibition of calcium ion accumulation, and hence reducing calpain-activated cell death mechanisms, could also improve photoreceptor survival in some forms of RP. The Ca^2+^ channel blocker Nilvadipine has been found to reduce photoreceptor loss in RCS rats and rd and rds mouse models of RP [172,173,174], presumably through a reduction in intracellular Ca^2+^ levels. Further, it was shown to slow the progression of some aspects of visual field loss in RP patients [175]. Nakazawa and colleagues observed significant improvement in average visual field sensitivity (central 10°). Central visual sensitivity (central 2°) improved but not significantly, with the authors suggesting this could be due to some of the control patients taking tocopherol or lutein, (not taken by any patients in the treated group) and the beneficial antioxidant effects of these medications could mask the significance of effects from Nilvadipine. They therefore speculate that a combination of Nilvadipine and antioxidant treatment may produce synergistic effects and lead to further improvement in photoreceptor survival. As with other studies, the genotypes of the patients were not known and this could also account for some of the variability in the results. This needs to be addressed in future studies to have any certainty that calcium channel blockers protect vision in RP.

### 6.2. Anti-Inflammatory Agents

As described in Section 4, chronic inflammation is often associated with the pathogenesis of RP. Intravitreal dexamethasone implants (Ozurdex) have recently been approved for the treatment of macular oedema related to retinal vein occlusion [176]. A recent small-scale study has also trialled use of dexamethasone implants for treatment of RP with associated cystoid macular oedema (CMO) [177]. They observed a reduction in CMO along with some restoration of retinal morphology with intravitreal steroid implants. Further, about 50% of subjects showed significantly improved visual acuity for up to four months post injection. However, repeated injections were required to maintain the effects, with the increased associated risk of cataract formation and raised intraocular pressure [177,178]. 

Rho-kinase (ROCK) inhibitors are able to suppress downstream effects of TNFα by decreasing intercellular adhesion molecule 1 (ICAM-1) expression and therefore reducing leukocyte adhesion to endothelial cells [179]. ROCK inhibitors such as Fasudil have been shown to reduce inflammatory cytokine expression and leukocyte-induced tissue damage to retinal and vascular epithelial cells in in vitro and in vivo studies [180]. Intravitreal injection of a ROCK inhibitor was also shown to improve photoreceptor function and reduce photoreceptor apoptosis in the RCS rat model of RP [181]. While ROCK inhibitors have yet to be tested in clinical trials for RP, intravitreal injection of Fasudil has recently been found to improve visual acuity in patients with diabetic macular oedema (DMO) when combined with a VEGF inhibitor, compared to anti-VEGF alone [182]. Again, the effects are short-term and repeated injections are required to sustain the beneficial effects. A longer-term delivery system would need to be developed to maintain the inhibition of inflammatory pathways without adverse effects. An inhibitor of C-C chemokine receptor 2/5 (CCR2/CCR5) has also been used in a clinical trial for treatment of DMO, resulting in a modest improvement in visual acuity, although the effect was not considered to be significantly greater than current anti-vascular treatments [183]. However, since macrophage recruitment appears to have a direct effect on photoreceptor loss in RP, beyond simply enhancing local inflammation [86], it is possible that cytokine receptor inhibitors could be more effective in treating RP.

### 6.3. Targeting ER Stress and Protein Homeostasis

One particular challenge is developing treatments for forms of RP resulting from dominant negative mutations that lead to the accumulation of mis-folded rhodopsin. It is necessary to remove the mis-folded protein and/or prevent the formation of aggregates as well as supplementing the level of wild type protein. Pharmacological interventions that improve protein folding or target the ER stress and UPR pathways are therefore being investigated. Small molecule chaperones have been shown to promote P23H rhodopsin folding and reduce cell death in vitro [184,185,186]. However, attempts to use these compounds to restore rhodopsin stability in vivo have produced conflicting results. For example, 4-PBA decreases aggregation of P23H rhodopsin in vitro [184] and reduces photoreceptor death in *Rho^P23H^* mice [37], yet systemic treatment with 4-PBA did not improve photoreceptor function or slow degeneration in *Rho^P23H^* rats [187]. Similarly, modulation of translation using the 5′AMP-activated protein kinase (AMPK) activator metformin improved P23H rhodopsin folding in culture, yet metformin treatment exacerbated rod cell death in *Rho^P23H^* rat and mouse models [188]. This study suggested that metformin-rescued P23H rhodopsin was still intrinsically unstable, but was trafficked to photoreceptor outer segments in larger amounts than wild type rhodopsin, leading to destabilisation of the outer segments. It is therefore not sufficient to correct mutant rhodopsin folding without improving stability, and the development of treatments has therefore focused on reducing aggregation or promoting degradation of mutant Rho. 

The chaperone network involving ER degradation-enhancing α-mannisidose-like protein 1 (EDEM1), ER-resident protein containing DNA-J 5 (ERdj5) and binding immunoglobulin protein (BiP) mediates ER quality control and has been shown to interact with mutant rhodopsin [189]. Over-expression of these chaperones in vitro reduces P23H Rho aggregation and promotes the degradation of mutant protein [189,190,191]. Over-expression of BiP was also found to reduce CHOP expression and photoreceptor death in *Rho^P23H^* rats; this was proposed to be due to suppression of ER stress-induced apoptosis rather than improved protein folding [192]. More recently, AAV-mediated over-expression of the BiP co-chaperone ERdj5 was shown to improve visual function and photoreceptor survival in *Rho^P23H^* rats [193]. Targeting components of this proteostasis network could therefore be an alternative means of treating RP resulting from mis-folded rhodopsin. However, as yet no clinical trials targeting these proteins have been published. 

Curcumin, a compound derived from turmeric, has been shown to have antioxidant and anti-apoptotic properties [194,195]. Curcumin was also found to inhibit the activation of microglia and release of chemokines in *rd1* mice, resulting in improved photoreceptor survival [107]. Treatment of COS-7 cells expressing P23H rhodopsin with curcumin resulted in the dissociation of protein aggregates and reduced levels of the ER stress markers BiP and CHOP [196]. This study also demonstrated that curcumin administration improved retinal morphology, rhodopsin localisation and photoreceptor function in *Rho^P23H^* rats compared to untreated controls. Although the precise mechanism by which curcumin reduces protein aggregation is not known, it has also been shown to inhibit aggregation of amyloid β and transthyretin, supporting the potential of curcumin as a treatment for degenerative diseases caused by mis-folded proteins [197]. A lecithin solid-state dispersion of curcumin has been tested in patients with diabetic retinopathy [198]. Although preliminary, the results of this study showed a reduction in DMO and improved visual acuity with few adverse effects, demonstrating the safety and suitability of curcumin for treating retinal degeneration. 

Tauroursodeoxycholic acid (TUDCA) inhibits the transport of Bax to the mitochondrial membrane, therefore inhibiting the initiation of apoptosis [199]. TUDCA may therefore reduce CHOP-induced apoptosis following ER stress. TUDCA has been shown to be neuroprotective in treatment of patients with amyotrophic lateral sclerosis [200] and preserved photoreceptor function in *rd10* and *rd1* mouse models of RP, Bardet–Biedl syndrome (*Bbs1*) mice [201,202,203] and *Rho^P23H^* rat models [187,204]. However, long-term administration of TUDCA did not result in the same protection of photoreceptor function in a *Rho^P23H^* mouse model [187]. The difference may be due to the balance of the pathways affected by different genetic causes; in this mouse model, P23H Rho accumulation leads predominantly to Ire1 up-regulation and ERAD [58], rather than the PERK and UPR activation observed in *Rho^P23H^* rats [205], and it is possible that this branch of the ER stress response is less influenced by TUDCA. More recently, however, Fernandez-Sanchez et al. developed a slow release delivery system for TUDCA using intravitreally injected biodegradable microspheres (TUDCA–polylactic-co-glycolic acid (TUDCA–PGLA)). They demonstrated that TUDCA–PGLA administration slowed photoreceptor loss in *Rho^P23H^* rats and increased ERG a- and b-wave amplitudes. There was also improved preservation of retinal structure as evidenced by immunolabelling with specific antibodies [206]. The authors propose that this form of slow-continuous delivery of TUDCA could provide a suitable therapy for at least some forms of RP, although the effects would need to be validated in other models. 

### 6.4. Novel Therapeutics for RP

In recent years, advances have been made in the use of gene augmentation therapy via AAV-mediated delivery vectors for the treatment of inherited retinal dystrophies (IRDs) [207,208,209]. As a result, we now have the technology in clinic to treat RPE65-mediated Leber’s congenital amaurosis. Further allele-specific treatments will likely follow. At the same time, the use of stem cell-derived cell replacement products for the treatment of IRDs has also reached clinical trial [210]. Although neither of these strategies directly address the cell death mechanisms of RP discussed in this review, we predict that in the future a combinatorial approach will be adopted, whereby drugs targeting cell death pathways will be used alongside gene and cell replacement, thus allowing RP patients to achieve their maximal visual potential.

## 7. Conclusions

Understanding the common mechanisms that lead to photoreceptor death in RP will be important for developing therapies that are effective across a range of underlying genetic causes. There is accumulating evidence for the involvement of non-apoptotic cell death pathways in RP, although the relative contribution of these pathways versus apoptosis is still to be determined. It is also possible that different cell death mechanisms are deployed depending on the causative mutation. Further research into the role of innate immune cells and the interplay between cell death pathways and autophagy will also be valuable for understanding the balance between protective and degenerative processes in RP.

## Figures and Tables

**Figure 1 genes-11-01120-f001:**
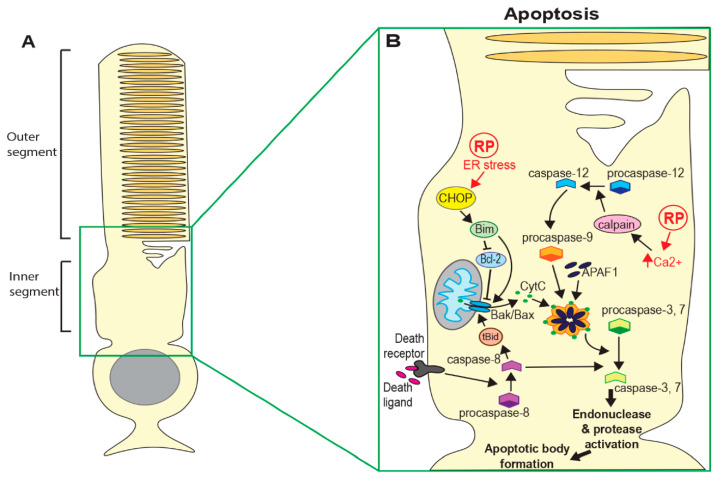
Cell death mechanisms in Retinitis pigmentosa (RP). (**A**) Schematic of a rod photoreceptor. (**B**) Both intrinsic pathways (through detection of cell damage) and extrinsic pathways (through interaction with immune cells and death receptors) can lead to apoptosis. Both pathways lead to the release of CytC from mitochondria and oligomerisation of CytC, APAF1 and pro-caspase-9 to form the apoptosome. Activated caspase-9 cleaves executioner caspases 3 and 7. (**C**) Photoreceptor death can occur by various forms of regulated necrosis. Necroptosis requires activation of RIP1/RIP3, leading to the activation and oligomerisation of MLKL, which forms pores in the cytoplasmic membrane. Parthanatos can occur following activation of PARP, which can be induced by high cGMP levels, leading to nuclear translocation of AIF. Some RP-causing mutations may lead to elevated Fe^2+^ and/or inhibition of GPX4, causing increased lipid oxidation and death by ferroptosis, although the mechanism is not known. (**D**) Macroautophagy can be initiated by endoplasmic reticulum (ER) stress via ATG12 activation, direct activation of ATG18/WIP12 and activation of beclin1. Phagophores engulf large cellular components and fuse with lysosomes to form autophagosomes and lysosomal enzymes then degrade the contents of autophagosomes. Proteins also enter lysosomes by hsc-70-assisted chaperone-mediated autophagy via the LAMP1 receptor and by microautophagy through invagination of the lysosomal membrane. (CytC = cytochrome C; APAF1 = apoptotic protease-activating factor 1; RIP = receptor-interacting serine/threonine-protein kinase; MLKL = mixed lineage kinase domain-like pseudokinase; PARP = poly-ADP ribose polymerase; AIF = apoptosis-inducing factor; GPX4 = glutathione peroxidase 4; ATG = autophagy-related protein; LAMP = lysosome-associated membrane glycoprotein).

**Figure 2 genes-11-01120-f002:**
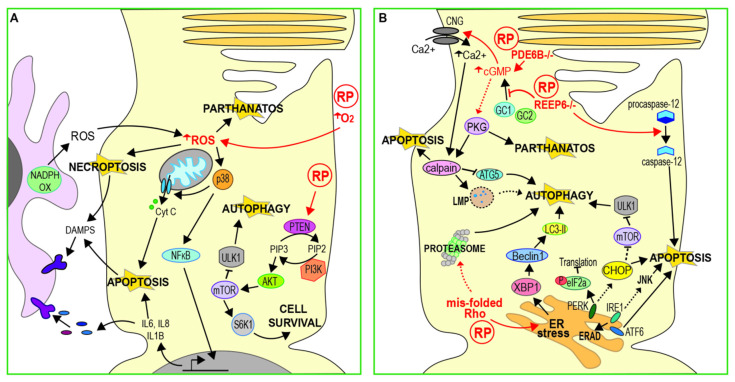
Pathways leading to rod photoreceptor death in RP. (**A**) Oxidative stress caused by increased extracellular O_2_ or extracellular ROS produced by activated microglia can induce apoptosis via the p38/NFkB pathway. Increased intracellular ROS may also trigger necroptosis or parthanatos. The mTOR pathway may be inhibited in RP through increased PTEN activity, this down-regulates cell survival pathways and increases autophagy. DAMPS released by apoptotic or necroptotic cells cause further activation of microglia. (**B**) Raised cGMP levels lead to increased Ca^2+^ influx, resulting in calpain activation and death by apoptosis. ER stress resulting from raised Ca^2+^ or accumulation of mis-folded rhodopsin activates PERK, Ire1 and ATF6. While this can lead to protective responses such as translation inhibition or ERAD, sustained ER stress can also trigger apoptosis. ER stress can also lead to increased autophagy and this, in turn, can lead to the degradation of proteasome subunits. Reducing autophagy allows more mis-folded rhodopsin to be degraded by the proteasome, thus reducing ER stress. Calpain activation can inhibit autophagy through lysosomal membrane permeablisation (LMP) and the cleavage of autophagy proteins such as ATG5. (ROS = reactive oxygen species; NFkB = necrosis factor k B; mTOR = mammalian target of rapamycin; ERAD = endoplasmic-reticulum-associated protein degradation; DAMP = damage-associated molecular patterns; PTEN = phosphatase and tensin homolog; PERK = protein kinase RNA-like endoplasmic reticulum kinase).

**Figure 3 genes-11-01120-f003:**
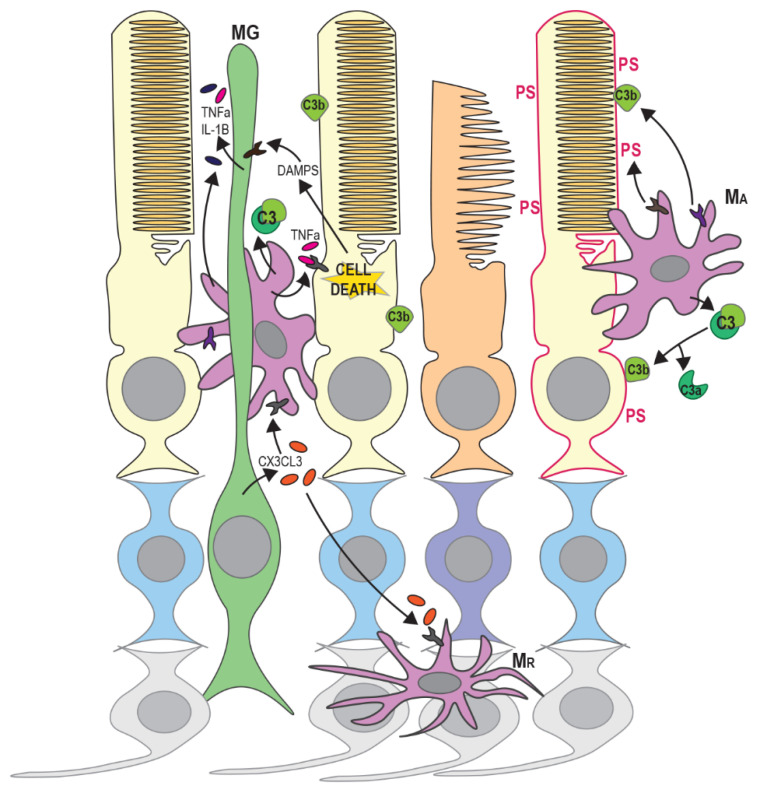
Microglia have an active role in photoreceptor death in RP. DAMPS released by apoptotic or necrotic photoreceptors trigger the release of inflammatory factors and cytokines from Müller glia (MG). Activated MG increase the expression and secretion of CX3CL3, leading to the activation of resting microglia (M_R_). Activated microglia (M_A_) secrete complement components including C3, which is cleaved to C3b on damaged or dying photoreceptors, marking these cells for phagocytosis. Stressed/damaged photoreceptors also expose phosphatidylserine (PS) on the outer surface of the cell membrane and this is also recognised by phagocytic receptors on activated microglia. The further release of inflammatory factors from M_A_ and MG induces apoptotic or necroptotic cell death in nearby photoreceptors. These processes affect rods (yellow) more than cones (orange) in early disease stages.

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
