# Peer review of "Mechanisms of Photoreceptor Death in Retinitis Pigmentosa"

_genes, 2020, doi:10.3390/genes11101120_

Round 1
Reviewer 1 Report
Dear authors,
in your review you are describing in detail and comprehensible the different mechanisms of cell death in RP, underlaying pathways and consequences regarding potential future treatment approaches.
The topic is of high interest and numerous research studies and clinical trials are ongoing to stop retinal degeneration or even develop regenerative therapies. Your review clearly points out the importance of 1) knowing the genotype of potentially treated patients and 2) knowledge about underlying degenerative mechanisms to avoid failure.
The review is well structured, does not miss key points and ends with a reasonable conclusion.
The figures are self-explaning and helpful to understand the complex interplay of involved pathways.
The only topic that could have been discussed in more detail are the Advanced Medicinal Therapies, i.e. gene and cell therapies. It is well understood that they are not the focus of this review; however, as already 30 studies are ongoing testing gene or cell treatment approaches, it seems worth to mention and summarize them briefly. There are 10 gene therapies (phase I/II) ongoing (aim to correct the RPGR gene, AAV-based) and 20 stem cell trials (up to phase II) registered at clinicaltrials.gov using MSC (12), human retinal and neural progenitor cells (4), hESC (2) or iPS cells (2).
Thus, I thank you for this excellent work and would only suggest a few more minor modifications:
- additional spelling check to correct some orthographic/grammar errors (e.e.g, page 14 line 536 "rapid in in mice"),
- Phosphatidylserine is mentioned first in figure 3, but explained later on page 13. Please introduce the abbreviation in the figure,
- the term "ameliorated progression of visual field loss" might be misleading due to the positively assessed word "amelioration" - increased/fastened or something similar might be more appropriate,
Author Response
Thank you for your review of our manuscript. We have addressed the points you raise as detailed below:
1) The only topic that could have been discussed in more detail are the Advanced Medicinal Therapies, i.e. gene and cell therapies. It is well understood that they are not the focus of this review; however, as already 30 studies are ongoing testing gene or cell treatment approaches, it seems worth to mention and summarize them briefly.
Although current gene and cell therapies do not address cell death mechanisms directly, so are not the focus of this review, we have added an extra paragraph (section 6.4) that briefly highlights some of the key trials employing these treatment approaches.
2) Phosphatidylserine is mentioned first in figure 3, but explained later on page 13. Please introduce the abbreviation in the figure.
The abbreviation is now defined in the legend of figure 3
3) the term "ameliorated progression of visual field loss" might be misleading due to the positively assessed word "amelioration" - increased/fastened or something similar might be more appropriate.
The work cited suggests oral supplements of DHA resulted in slower progression of visual field loss over the time course of the study. The word 'ameliorated' (line 645) has therefore been changed to 'slowed' to make the meaning clearer.
4) correct some orthographic/grammar errors (e.e.g, page 14 line 536 "rapid in in mice")
Orthographic errors have been corrected.
Reviewer 2 Report
This work seeks to extensively review the current state of the research on the molecular mechanisms that lead to photoreceptor death in retinitis pigmentosa, and the emerging therapies to treat this inherited retinal dystrophy. The manuscript is in general well written and clearly summarises and discusses the main findings on the field. Moreover, the authors also expose the recent clinical trials developing pharmacological approaches to target photoreceptor cell death.
It is worth noting that this review presents a lot of information related to concrete cell death mechanisms and subcellular pathways taking place in RP photoreceptors, and thus graphic support stands essential for the correct understanding of the content. In this sense, my only concern relates to the figures of the manuscript, which in my opinion present repetitive information and are unbalanced: figures 1 and 2 contain many cellular pathways and processes (some of which redundant), whereas figure 3 is in comparison poorly informative. For this reason, I recommend rearranging the three figures of the present manuscript in order to better illustrate and support the information on the text.
In addition, other minor modifications should be addressed within the text:
- Lines 143, 205, 574, 749 and 752. Regarding the genetic basis of disease, the authors usually refer to as "mutation". While it is not strictly incorrect, the term is not precise enough. For this reason, I recommend using "genetic cause" instead, to include not only the concept of mutation but also the implicit gene altered.
- Line 249. "knowns" should be corrected to "known".
- Line 282. "gualylate" should be replaced by "guanylate".
- Line 536. "in" is repeated.
- Line 598. "received" is repeated in the same sentence.
Author Response
Thank you for your review of our manuscript. We have addressed the points you raise as detailed below:
1) My only concern relates to the figures of the manuscript, which in my opinion present repetitive information and are unbalanced: figures 1 and 2 contain many cellular pathways and processes (some of which redundant), whereas figure 3 is in comparison poorly informative. For this reason, I recommend rearranging the three figures of the present manuscript in order to better illustrate and support the information on the text.
Figure 3 describes the known mechanisms of Muller glia/ microglia activation and recruitment. Since there is not as much literature describing this as for other cell death mechanisms, this figure will inevitably be less detailed and informative than figures 1 & 2. We cannot add any more to this figure. Regarding the redundancy between figures 1 & 2: figure 1 describes known mechanisms of general cell death, as outlined in section 2 (Cell death mechanisms), whereas figure 2 attempts to represent graphically the evidence for pathways in RP that lead to these in the photoreceptor. Thus there will be some overlap between them. However, we have revised figure 2 to remove some elements that are already shown in figure 1 (induction of apoptosis by calpains, for example) and combined 2b and 2c into a single panel to remove any repetition/ redundancy between them.
2) Regarding the genetic basis of disease, the authors usually refer to as "mutation". While it is not strictly incorrect, the term is not precise enough. For this reason, I recommend using "genetic cause" instead, to include not only the concept of mutation but also the implicit gene altered.
Lines 143, 208, 547, 785, 759 'mutation' has been replaced with 'genetic cause'.
3)
- Line 249. "knowns" should be corrected to "known".
- Line 282. "gualylate" should be replaced by "guanylate".
- Line 536. "in" is repeated.
- Line 598. "received" is repeated in the same sentence.
These errors have been corrected (new line numbers: 257, 290, 554, 617).